# Robotically Assisted Surgery in Children—A Perspective

**DOI:** 10.3390/children9060839

**Published:** 2022-06-06

**Authors:** Thomas Franz Krebs, Isabel Schnorr, Pascal Heye, Franz-Martin Häcker

**Affiliations:** 1Department of General-, Visceral-, Thoracic-, Transplant- and Pediatric Surgery, UKSH University Hospital of Schleswig-Holstein Kiel Campus, Arnold-Heller-Strasse 3, 24105 Kiel, Germany; 2Clinic for Pediatric Surgery, Children’s Hospital of Eastern Switzerland, Claudiusstrasse 6, 9006 St. Gallen, Switzerland; pascal.heye@kispisg.ch (P.H.); frank-martin.haecker@kispisg.ch (F.-M.H.); 3Faculty of Medicine, University Hospital Frankfurt-Goethe University, Theodor-Stern-Kai 7, 60590 Frankfurt am Main, Germany; isabel.schnorr@kgu.de

**Keywords:** robotically assisted surgery, pediatric robotic surgery, minimally invasive surgery, robotics, 3 mm instruments, microlaparoscopy, 3 mm RAS instruments

## Abstract

The introduction of robotically assisted surgery was a milestone for minimally invasive surgery in the 21st century. Currently, there are two CE-approved robotically assisted surgery systems for use and development in pediatrics. Specifically, tremor filtration and optimal visualization are approaches which can have enormous benefits for procedures in small bodies. Robotically assisted surgery in children might have advantages compared to laparoscopic or open approaches. This review focuses on the research literature regarding robotically assisted surgery that has been published within the past decade. A literature search was conducted to identify studies comparing robotically assisted surgery with laparoscopic and open approaches. While reported applications in urology were the most cited, three other fields (gynecology, general surgery, and “others”) were also identified. In total, 36 of the publications reviewed suggested that robotically assisted surgery was a good alternative for pediatric procedures. After several years of experience of this surgery, a strong learning curve was evident in the literature. However, some authors have highlighted limitations, such as high cost and a limited spectrum of small-sized instruments. The recent introduction of reusable 3 mm instruments to the market might help to overcome these limitations. In the future, it can be anticipated that there will be a broader range of applications for robotically assisted surgery in selected pediatric surgeries, especially as surgical skills continue to improve and further system innovations emerge.

## 1. Introduction

The introduction of robotically assisted surgery (RAS) was a milestone for minimally invasive surgery in the 21st century. Previously, laparoscopic approaches (LA) represented the first step away from open surgery arising from new technical innovations [1]. In 2000, the da Vinci^®^ surgery system (Intuitive Surgical, Sunnyvale, CA, USA) revolutionized the industry by applying principles that were similar to those of LA but that used remotely controlled instruments [2]. This RAS system was the first to be approved by the CE, with initial surgeries using the system being performed and evaluated in 2001 [3]. In 2020, a second robotic system became CE-approved for pediatric surgery, the Senhance^®^ surgical system (Asensus Surgical, Durham, NC, USA) [4]. Building on what was learned from the limitations of the da Vinci^®^ surgery system, the system included several technical improvements.

As early as 2004, Woo and colleagues, in their review, had already indicated the potential of RAS for pediatric surgery [5]. Given children’s smaller anatomical structures, it has been suggested that the advantages of RAS include tremor filtration ability and a greater range of motion [6,7,8,9]. Additionally, technologies such as eye-tracking cameras and three-dimensional visualization are now available for optimal spatial vision and precise handling. As an illustration, early studies using RAS in piglets showed that sutures could be performed safely in cavities as small as 92 mL [10].

As in the adult population, RAS in children is mostly applied in the field of urology [11]. Nevertheless, it is reasonable to extend the areas of application of RAS given its advantages. With respect to other surgical specialties, reported comparisons between RAS and LA and open approaches are sparse and ambiguous for many procedures. Therefore, this review seeks to consider and summarize recent outcomes of comparison studies and to explore whether RAS represents a valid alternative across different surgical procedures.

The authors do not intend to address the specific outcome comparisons for individual procedures. Instead, the review aims to present current evidence supporting RAS in context and to highlight further directions for future research.

## 2. Materials and Methods

To conduct the review, electronic databases, such as MEDLINE^®^/PubMed, Cochrane, World of Knowledge, and Google Scholar, were searched independently to identify eligible literature evaluating RAS compared to laparoscopic and open approaches in pediatric surgery. The search terms used included “robotically assisted”, “robotic”, “surgery”, “children”, “pediatric”, “urology”, “general surgery”, “gynecology”, “laparoscopic”, “open”, and “versus” in various combinations with the help of “AND” and “OR”. The inclusion and exclusion criteria were discussed and approved by all the authors. Title, abstract, and full-text screening, as well as the assessment of article eligibility, were performed by the authors TK and IS independently in February 2022. Where there were disagreements, these were discussed with a third author. If available, emphasis was placed on major studies, such as meta-analyses and systematic reviews. Recent literature published between 2013 and 2022 was preferentially considered. With respect to demographics, the pediatric patients in study samples were required to have an average weight of at least 10 kg or to be at least one year of age for inclusion, but no further restrictions were applied. These selection criteria were chosen because RAS is technically more challenging in newborns and infants (age < 1 year; weight < 10 kg). The smaller working space increases the risk of complications, such as robotic arm collisions or suboptimal trocar placement [6,12]. Additionally, the conclusions (e.g., on safety) that have been reported in the data published for this subgroup are sparse [6,13]. It was required that the reported surgery be abdominal or thoracic. Data from other surgical specialties (e.g., orthopedic surgery or neurosurgery) were not considered. Further, the selected publications were required to provide outcomes that explicitly compared RAS to laparoscopic or open approaches in terms of the operating time, success rate, complication rate, conversions, length of hospital stay, reintervention, costs, and learning curves, or to provide conclusions that enabled an indirect comparison. A PRISMA flow diagram is provided in the Appendix A (Appendix A).

## 3. Results

After removal of duplicates, 50,102 titles and 1073 abstracts of unique articles were screened. A total of 433 full-text articles were assessed for eligibility, and 397 were excluded for not meeting the inclusion criteria. In total, 36 papers were finally included for detailed review. The use of RAS was most frequently reported for the following surgical specialties: urology, gynecology, general surgery, and “others”. The category “others“ included surgical oncology and cardiac surgery, or summarized the evaluation of several procedures.

The use of RAS in children is still developing. Therefore, a lack of data, inadequate study designs, and insufficient statistical methods dominated the investigated publications. For this reason, this review is constrained in presenting differentiated outcomes. The results that are presented address variables such as operating time, success, complications, conversions, length of hospital stay, reintervention, costs, and learning curves per procedure, as well as indications for the use of RAS in particular surgical specialties.

### 3.1. Pediatric Urology

Since its first implementation in pediatric surgery, an extensive amount of literature has developed on the use of RAS for urologic interventions. The most common procedures undertaken are pyeloplasties and ureteral reimplantations [11].

#### 3.1.1. Pyeloplasty

Pyeloplasty is one of the most common types of RAS in pediatric urology for typical congenital ureteral anomalies, such as ureteropelvic junction obstruction (UPJO). As a result, numerous meta-analyses and systematic reviews regarding this procedure are available.

Greenwald and colleagues recently published a systematic review and meta-analysis of 58 observational and 46 retrospective studies evaluating the use of RAS for pediatric pyeloplasty [14]. They included studies published between 2005 and July 2020 on children with an average age of 8.2 years. The study found a success rate of 95.4% and an average complication rate of 12%. Even though they did not directly compare RAS to LA or open approaches, the authors assessed the success rate to be similar to that of open pyeloplasty. Additionally, a shorter hospital stay (~1 day) and shorter operating time than open approaches resulted in favorable outcomes [14]. The authors concluded that RAS pyeloplasty was feasible and safe. However, they criticized the high levels of heterogeneity in the definitions and the reporting of successes, complications, operating times, and safety. To provide an example, data on Clavien-Dindo gradings were hardly stated, making it difficult to derive general conclusions on post-operative complications.

Another meta-analysis compared RAS and LA pyeloplasty [15]. This study looked at 14 observational studies that included a total of 2254 children. Overall, the results of the study showed a significantly shorter hospital stay for RAS than for LA pyeloplasty (mean difference: −1.49; 95% CI −2.22, −0.77, *p*-value < 0.0001). RAS also resulted in a higher success rate (RAS: 98.2%, LA: 96.2%). No significant differences were observed in the operating time, post-operative complication rate, or reintervention between the two approaches. This meta-analysis suggested that RAS was safe, with higher success rates than LA pyeloplasty. However, the authors discussed the need for more randomized controlled trials to assure the reliability of the study’s conclusions.

The prevalence of recurrent UPJO can be up to 11.5% in pediatric surgery and requires a reintervention [16]. Reinterventions on recurrent UPJO might be more challenging than the initial procedure, so it seems reasonable to investigate the potential of RAS further.

Addressing redo-pyeloplasty and its outcomes, 40 studies, involving a total of 1549 children aged between 24 and 205 months old, were reviewed showing a pooled conversion rate of 1.5% from RAS pyeloplasty to LA or an open approach [17]. Concerning postoperative complications, an 11.7% frequency of incidents occurred for RAS, with the majority represented by Clavien I complications (fever, pain, and haematuria). The most observed complications for Clavien complications ≥ II were urinary tract infections. More specifically, 13.5% of Clavien IIIb complications involved stent migrations that needed reintervention. The success rate was between 84% and 100%, with a tendency for success greater than 90% in younger and lighter children. Masieri and colleagues pointed out the technical challenges and poor maneuverability of LA pyeloplasty, especially during the reconstructive phase. These challenges could cause LA pyeloplasty to have longer operating times than RA (LA (median range): 67.8–384 min). In this meta-analysis, the advantages of RAS over LA and open approaches were highlighted in challenging intraoperative scenarios, with RAS shown to produce safe and beneficial outcomes.

In 2021, a large retrospective study compared a sample of 276 primary with 30 redo RAS pyeloplasty operations [18]. In children with an average age of 4.9 years at the time of surgery, the redo RAS had longer operating times (mean primary: 198.9 min, mean redo: 278.0 min, *p*-value = < 0.001) and longer hospital stays (mean primary: 1.3 days, mean redo: 2.0 days, *p*-value = <0.001) than the primary RAS. However, no significant differences were observed regarding post-operative complications (primary: 13.0%, redo: 16.7%, *p*-value = 0.58) or the need for additional procedures (primary: 4.3%, redo: 6.7%, *p*-value = 0.56). The authors summarized their study results for redo RAS pyeloplasty, indicating that it had low complication and high success rates (primary: 95.3%, redo: 90%, *p*-value = 0.20) that were comparable to those of primary RAS pyeloplasty. Together, the findings of this study indicated that redo RAS pyeloplasty was an efficient and safe approach for the reconstruction of recurrent UPJO.

#### 3.1.2. Ureteral Reimplantation

Ureteral reimplantation (UR) is another common procedure in pediatric urology. It is typically performed to treat vesicoureteral reflux. Here, the application of RAS is well-established, and a large number of studies have been published.

A systematic review exploring RAS for UR in 1362 children with an average age of 5.4 years was recently published [19]. The study found rates of 92% success, 1.5% intraoperative complications, 10.7% postoperative complications (8.2%: Clavien I-II, 2.5%: Clavien III), and 3.9% reinterventions. The authors explicitly mentioned that there was no correlation between the success rates and complex anatomy or previous procedures. Therefore, challenging procedures using RAS were not associated with alterations in complications. The study compared RAS to open approaches and indicated longer operating times for RAS but no difference in the length of hospital stays (*p*-value = 1.00) or complication rates (*p*-value = 0.32). Compared to RAS, LA had a tendency for shorter operating times (*p*-value = 0.1003) and no differences in terms of complications and success rates (*p*-value = 0.9163). Of note, the authors reviewed early (2008–2016) versus more recent (2017–2019) publications. More recent papers showed greater success (92.8%) and lower failure rates (5.2%) compared to earlier papers (90.9% and 9.2%, respectively). In addition, fewer post-operative complications (Clavien 1-3) and less need for reintervention were reported in the more recent papers (*p*-value = 0.001). In general, the more recent papers indicated that RAS resulted in improved surgical outcomes. Importantly, the learning curves of surgeons when learning how to carry out RAS were found to be similar to those of other new surgical procedures, with no considerable challenges being observed. This conclusion can be applied not only to UR but also to other procedures such as pyeloplasties. Esposito and colleagues determined RAS to be feasible and effective and suggested that it could be a first-line surgical approach for vesicoureteral reflux.

In 2018, Deng and colleagues investigated RAS UR in 7122 children [20]. The study directly compared RAS to open approaches performed between 2003 and 2014. Here, RAS was associated with longer operating times (mean difference: 66.69 min; 95% CI 41.71–91.67; *p*-value = 0.00001) but resulted in shorter hospital stays (mean difference: 17.80 h, *p*-value ≤ 0.00001). No significant differences were found regarding success, intraoperative blood loss or complication rates. However, RAS showed more short-term postoperative complications (hydronephrosis, urinary tract infection, and hematuria) than open approaches. Based on two study results, the authors also referred to the significantly higher cost of RAS compared to open approaches (*p*-value = 0.001 and *p*-value = 0.049). It was argued that RAS UR is an effective surgical approach for pediatric UR but that more randomized controlled studies are needed.

Another review comparing RAS and LA UR included 28 papers that were published between 2001 and 2020 [21]. The pediatric patients included in this study had an average age of 5.8 years. In contrast to previous publications, this study indicated that for UR, RAS had lower success rates (RAS: 93.4%, LA: 97.6%, *p*-value = 0.0018), longer operating times for uni- and bilateral procedures (mean RAS: 171/223 min, mean LA: 107/161 min, *p*-value ≤ 0.001), and longer hospital stays (mean RAS: 1.8 days, mean LA: 1.6 days, *p*-value = 0.002). However, no differences were detected in complication rates (*p*-value = 0.32). The outcomes of the systematic reviews and meta-analyses undertaken suggest that RAS is a good alternative to open approaches for UR and that an improved learning curve may enable RAS UR application to catch up with established approches. As highlighted in previous studies, the authors strongly recommend significant cost reductions for robotic systems, smaller instruments, and standardized study outcome reporting.

#### 3.1.3. Partial Nephrectomy

Partial nephrectomy is a less common RAS in children that is used to remove atrophic or multicystic dysplastic kidneys.

In a wide-ranging review, Andolfi and colleagues suggest the technical feasibility of using RAS nephrectomy in selected pediatric patients and address the benefits of performing multiple surgeries simultaneously, such as nephrectomy with contralateral ureteral reimplantation [22]. The study also considered RAS’ better dexterity for the prevention of vascular accidents (4–5%). The authors of the review stated that the use of RAS was feasible, but that its advantages over other approaches were questionable.

In a meta-analysis consisting of 22 papers, Grivas and colleagues evaluated the application of RAS for partial nephrectomy compared to open approaches [23]. RAS provided better results with respect to estimated blood loss and length of hospital stay, and showed similar complication rates compared to open approaches. The authors concluded that RAS was safe and effective even for highly complex procedures.

#### 3.1.4. Others

As RAS implementation has continued to be introduced in pediatric surgery, it has become possible to assess the feasibility of its application for a wider range of clinical indications. The positive outcomes of RAS have resulted in its application in other complex procedures, such as in bladder neck surgery [24], urachal remnant excision [25], and urolithiasis [26,27]. Famakinwa and Gundeti reviewed RAS for Mitrofanoff appendicovescicostomy, another urological reconstructive procedure in children [28]. Compared to open approaches, the study found a shorter hospital stay (median RAS: 5 days, median open: 8 days) and less intra- and post-operative pain (i.e., morphine usage: RAS: 0.88 and 0.68 mg/kg, open: 1.2 and 1.0 mg/kg), respectively, while RAS had longer operating times (RAS: 323 min, open: 267 min). This study demonstrated that RAS was feasible with equivalent effectiveness, morbidity, and complications. Additionally, it was also able to handle fine anatomical structures more delicately than other approaches. The authors emphasized the need for development and implementation of smaller instruments as a direction for future research.

### 3.2. Pediatric Gynecology

There have been some minor applications of RAS in pediatric gynecology. The lack of meta-analyses and systematic reviews makes it difficult to draw conclusions regarding the current use of RAS in this field. However, there are some available reports that should be considered.

One study provided a report on four children with an average age of 7.5 years and a mean weight of 36.8 kg [29]. In 2017, the authors successfully performed RAS based on clinical indications of two ovarian mature cystic teratomas, one mucinous tumor of the ovary, and one ovarian teratoma. Despite limited experience, the authors still found RAS to be simple, safe, and effective for selected pediatric patients. The authors considered the resection of ovarian tumors (oophorectomy) to be a suitable introductory procedure for surgeons who are new to RAS.

Almost ten years ago, a related small case series was reported with six pediatric patients aged between 2.4 and 15 years and weighing between 12 and 55 kg [30]. Nakib and colleagues used RAS for two ovarian cystectomies, two oophorectomies, one right oophorectomy, one left salpingo-oophorectomy for gonadal dysgenesis, and one exploration for suspected pelvic malformation. The study showed a mean operating time of 117.5 min, no conversion to LA and no intra- or post-operative complications. The authors interpreted their first results in the use of RAS in pediatric gynecological procedures as indicating that the method was safe. However, they were not able to confirm improved clinical outcomes compared to LA. As suggested by others, the authors criticized the high cost of RAS and expressed the need for randomized controlled studies.

Kebodeaux and colleagues successfully performed the excision of Müllerian remnants, reflecting their complex anatomy and surgical history, in two girls aged 13 and 16 years [31]. The authors found that RAS enabled an improved range of motion for optimal handling in complex anatomical conditions. Their findings demonstrated that the use of RAS was feasible for this procedure.

Similarly, another study reported on the excision of Müllerian remnants in three children [32]. For this procedure, RAS improved the visualization and precision required for challenging dissections in the deep pelvic region, suggesting a lower risk of injuring surrounding structures. Summarizing their first experiences, the authors considered RAS to be safe and effective.

Finally, a review by Pelizzo and colleagues suggested the use of RAS for additional procedures, such as for the excision of adnexal masses and uterine horn remnants, due to the improved visualization in narrow spaces that it offered [33].

### 3.3. Pediatric General Surgery

RAS has also been used in pediatric general surgery; however, to date, it has not reached “state of the art” status compared to LA or open approaches.

#### 3.3.1. Cholecystectomy and Resection of Choledochal Cysts

The removal of the gallbladder or anatomically related cysts is a standard procedure in pediatric general surgery and represents a field where RAS may be helpful.

A study using data collected from January 2015 to December 2018 involved 299 cases of cholecystectomy in children with an average age of 15.5 years [34]. A total of 220 pediatric patients underwent LA and 79 underwent RAS. Notably, most of the children who underwent RAS were in non-acute situations, while surgical interventions for acute indications (e.g., acute cholecystitis) were performed with LA. The operating time was longer in RAS (mean RAS: 98 min, mean LA: 79 min, *p*-value ≤ 0.001), but no significant differences were observed in terms of post-operative complications. Unlike most other studies, the authors compared their costs per case; the average total hospital costs for RAS were USD 15,519 compared to USD 11,197 for LA. Thus, the costs for RAS were significantly higher. Apart from the higher costs and longer operating time, this study concluded that RAS was feasible for cholecystectomy and that it had similar outcomes compared to LA.

Another study systematically reviewed RAS choledochocystectomy from eight publications including 86 pediatric patients with an average age of 6.3 years [35]. Compared to open approaches, the study showed a conversion rate of 8.1%, while the success rate was 91.9%. The documented operating time for the procedure was 426 min on average. Based on their findings, the authors concluded that RAS was safe and feasible. Similar to other reports, they emphasized the advantages of RAS over LA in terms of more flexible and accurate surgery resulting in smaller injuries, enhanced visualization, and precise manipulation.

The largest analysis to date focusing on the use of RAS for the resection of choledochal cysts was recently published by Ihn and colleagues [36]. The authors published their data on 158 children who were all older than 3.7 years, based on a single-center study. The procedures evaluated were divided into a first period (P1 = July 2008–March 2016, N = 79) and a second period (P2 = April 2016–January 2021, N = 79), with the second period showing significantly shorter operating times (mean P1: 462.6 min, mean P2: 383.6 min, *p*-value ≤ 0.001) and less blood loss (P1: 63 mL, P2: 28 mL, *p*-value = 0.025). The two approaches showed no significant differences concerning conversions (one conversion in each period), post-operative complications, hospital stays, or reintervention rates. This study provided evidence of successful learning and improved RAS skills, and indicated that RAS involved safe and feasible procedures.

#### 3.3.2. Gastric Fundoplication

Fundoplication is already well-established in robotically assisted anti-reflux surgery for the surgical treatment of gastroesophageal reflux disease.

A study compared the outcomes of 150 Nissen cases consisting of RAS (N = 50), LA (N = 50), and open approaches (N = 50) [37]. Only studies that were completed up until 2005 were included. The children undergoing fundoplication had an average age of 117 months. In this study, RAS was associated with longer operating times (mean RAS: 160 min, mean LA: 107 min, mean open: 73 min, *p*-value = 0.05) and had a similar conversion rate to open approaches (RAS: 4%, LA: 2%). Regarding hospitalization, open approaches were associated with longer hospital stays compared to RAS (mean RAS: 2.94 days, open: 3.5 days, *p*-value ≤ 0.05). There were no differences in the complication rates for adverse events, such as hiatal hernia and wound infection between the different approaches (RAS = 14%, LA: 8%, open: 10%, *p*-value = 0.387). A large number of published studies do not report follow-up information; this study, however, provided documented information on 30-day follow-up symptoms (dysphagia, abdominal pain, feeding aversion, and gas bloating) and found these to be similar among approaches (RAS: 30%, LAS: 28%; open: 12%, *p*-value = 0.06). The study concluded that RAS was equivalent to LA in terms of complications and length of hospital stays. Addressing the longer operating time, the authors suggested that further experience with RAS was needed to overcome the learning curve and to reduce the time taken. 

Binet and colleagues examined sixty robot-assisted fundoplications [38]. The authors divided their cases into three periods: period I was represented by cases 1–15, period II was represented by cases 16–30, and period III was represented by cases 31+. The documented docking time decreased steadily from 12 min (period I), to over 5 min (period II), to 4 min (period III) (*p*-value = 0.004). Similarly, the operating time was also reduced, on average, from 154 min (period I), to over 131 min (period II), to 80 min (period III) (*p*-value = 0.025). The study reported no conversions to an open approach in any period but presented one intraoperative complication for period I and post-operative complications for period I and period II (one complication for each period). The hospital stays were equivalent over all three periods (*p*-value = 0.199). In conclusion, the authors of this study suggested that continuous learning improved RAS outcomes even after 30 cases. Moreover, they suggested that RAS had certain advantages, such as potential benefits in terms of operating time and shorter technical training periods. This study’s findings indicated that RAS was safe, feasible, and represented an equally effective alternative to LA.

#### 3.3.3. Esophagogastric Dissociation

Another anti-reflux intervention and alternative to fundoplication is represented by esophagogastric dissociation. This procedure is applied particularly in neurologically impaired children, who often suffer from neuromuscular incoordination and severe gastroesophageal reflux [39]. However, in these cases, surgical treatment has a high failure rate and often requires reintervention [40]. Therefore, the potential of RAS deserves to be examined further for this procedure.

A small study compared total esophagogastric dissociation in five RAS and five LA cases [41]. RAS was associated with longer operating times (median RAS: 290 min, median LA: 170 min, *p*-value = 0.004) and increased 7-day postoperative complications. Four complications arose in the RAS group compared to none in the LA group. No other differences between the approaches, such as length of hospital stay (RAS: 17 days, LA: 18 days), or short-term complications (*p*-value = 1.00), were reported. Furthermore, none of the procedures showed intraoperative complications or required conversion to open surgery. The authors stressed the benefits of comfortable and more accessible working conditions with RAS, especially when considering the dystonic posture of neurologically impaired children. This small study concluded that the results obtained with RAS were similar to those for LA, and, thus, recognized RAS as feasible.

#### 3.3.4. Splenectomy

The removal of the spleen is a procedure required for several hematologic or infectious disorders, trauma, and anatomical abnormalities. Whenever it is available and feasible, LA is already preferred over open approaches, though the literature already includes reports of several applications of RAS.

In 2020, a direct comparison between 10 RAS and 14 LA splenectomies for hematologic disorders was carried out [42]. Children underwent surgery between 2014 and 2019 and were on average at least 9.9 years old. The study results showed that the operating time was similar between both approaches (RAS: 140.5 min, LA: 154.9 min), and the need for intraoperative blood transfusions, associated transfusion volumes, postoperative complications, as well as 30-day outcomes, were not significantly different. Intraoperative conversion to an open approach was recorded for one RAS case due to intraoperative bleeding. However, the median hospital stay was shorter for RAS than LS (RAS: 2.1 days, LA: 3.2 days, *p*-value = 0.02). With respect to costs, the median total charges were significantly higher for RAS compared to LA (RAS: USD 44,724, LA: USD 30,255, *p*-value = 0.01). In their discussion, the authors concluded that RAS was safe and feasible with comparable operating times and post-operative morbidity, decreased length of stay, but higher charges. Interestingly, the authors also detailed references that suggest that RAS may decrease the cognitive stress experienced by surgeons.

Another single institution report involved 32 RAS and 23 LA splenectomies, with an age range of the children of 2 to 18 years [43]. Over a period of 11 years (2003–2014), on the one hand, the report demonstrated, in line with a previous study [42], shorter mean operating times for RAS than for LA (RAS: 159.6 min, LA: 182.4 min), but, on the other hand, longer hospital stays for RAS than for LA (RAS: 3.93 days, LA 2.9 days). One case for RAS was converted to an open approach but no cases were for LA. Investigations of post-operative complications revealed two surgical complications (e.g., reoperation for hemoperitoneum) for LA and six medical complications (e.g., atelectasis, pneumonia, sickle cell crisis, and generalized feeling of malaise) for RAS. The authors interpreted the findings to indicate insignificant differences in post-operative complications. Finally, the authors of this report highlighted important concerns, such as the adaptability of robotic instruments to pediatric patients’ smaller bodies and anatomical structures, the generally high costs, as well as the time required for setup installation.

#### 3.3.5. Congenital Inguinal Hernia

Although congenital inguinal hernias are probably the most frequent indication for pediatric surgery, herniotomies are not necessarily suitable for RAS and LA due to the excellent results and low cost of open approaches. Only a few reports have focused on RAS for inguinal hernias.

A small case series examined 11 robotically assisted inguinal hernia surgeries. All children were male and had an average age of 17 years and an average weight of 76.6 kg [44]. The average operating time was 111 min with minor blood loss (estimated < 5 mL). RAS resulted in no intraoperative complications, and no conversion to an open approach was needed. In this study, the children were discharged on the same day of the procedure. Within six months and four years of follow-up, no post-operative complications, such as infection or hernia recurrence, occurred. The study outcomes suggested that RAS is a safe and reliable approach for inguinal hernia in adolescents.

#### 3.3.6. Hirschsprung

Hirschsprung is a congenital disease that affects the distal colon and results in functional obstruction. Children who are affected by this disease are relatively young at diagnosis, and investigations on RAS to treat this malformation seem necessary.

In a retrospective study, eleven pediatric patients with an average age of 29 months underwent RAS for Hirschsprung [45]. The study found a median operating time of 420 min and no intraoperative complications. Prato and colleagues state that the use of RAS for Hirschsprung is appropriate, even in those pediatric patients that require a redo intervention. They recognized RAS as a valid alternative to other available approaches.

More recently, Quynh and colleagues reviewed data for December 2014 to December 2017 relating to procedures for Hirschprung treatment [46]. They considered 55 pediatric patients with an average age of 24.5 months. The study showed no conversion to LA, open approaches, or intraoperative complications. The hospital stay was, on average, 5.5 days, and follow-up at 43.2 months (median) after surgery showed good outcomes for the procedure. The authors found RAS to be a safe and effective technique for Hirschsprung treatment. However, they pointed out that skilled teams are essential, and that smaller angulated instruments and cost reduction are needed. The authors agree with the authors of other studies in indicating that studies on long-term outcomes are urgently required to confirm any superiority of RAS over LA.

### 3.4. Others

#### 3.4.1. Pediatric Oncology

The most extensive prospective series of RAS in pediatric oncology was published in 2021 [47]. Between 2016 and 2020, 100 tumors were operated on in 89 children. A total of 77% of these tumors were abdominal (67%) or pelvic (10%), with the most common procedure being adrenalectomy (N = 28). An 8% conversion rate to an open approach was reported. Interestingly, 71% of these cases occurred in procedures for renal tumors. Four children showed five post-operative complications (Clavien III) that required reintervention within the first 30 days after surgery. The authors concluded that RAS was a safe option in highly selected cases and emphasized the benefit of improved possibilities for complex resection in pediatric oncology compared to LA.

#### 3.4.2. Pediatric Cardiac Surgery

To date, cardiac surgery for congenital intracardiac pathologies has seldom been performed with LA or with RAS. Reports on the implementation of RAS are therefore sparse.

A study investigated 30 pediatric patients who underwent RAS from May 2013 to June 2018 for the indications of total atrial septal defect closure (N = 22), right-sided (N = 5) or left-sided (N = 1) partial anomalous pulmonary venous connection repair, tricuspid valve annuloplasty (N = 4), and mitral valve replacement (N = 2) [48]. The mean age of the study population was 16.1 years. The operating time was, on average, 4.1 h, with no conversions or complications. After the procedures, the pediatric patients stayed in the hospital, on average, for 3.3 days. Overall, it was concluded that RAS was feasible and safe for cardiac surgery. Nevertheless, the authors suggested that RAS should only be used in selected pediatric patients. They emphasized that, in the thorax especially, narrow conditions cannot be enlarged by capnoperitoneum, as they can in the abdomen, and therefore, the development of smaller RAS instruments is required.

#### 3.4.3. Others

A study by Lima and colleagues examined 40 children undergoing various procedures carried out with RAS (13 gastrointestinal, 18 urological, 7 thoracic, and 2 gynecological procedures) compared to 112 children who underwent conventional minimally invasive surgeries (i.e., LA) [49]. In the RAS group, the children were, on average, 12 years of age. Those undergoing LA procedures were younger, with an average age of 7.5 years. The pediatric surgeries were not identical for the groups; however, RAS was associated with significantly longer operating times (RAS: 116.8 min, LA: 80.1 min). The conversion rate to open approaches was similar between both groups: 2.5% in RAS and 1.8% in LA (*p*-value ≥ 0.05). The hospital stay was 6.1 days for RAS but not significantly lower for LA (5.3 days, *p*-value = 0.4). The study results indicated comparable safety and feasibility. The significant potential of RAS, including its intuitive handling and high maneuverability, were emphasized for complex procedures and narrow conditions. Especially in smaller children, the authors noted limitations associated with instrument size and the need for miniaturization to improve safety.

Another large study reported on 539 children who underwent 601 procedures [50]. In total, 45 different types of procedures were performed. The overall conversion rate was found to be 5.8%; the rate was significantly higher in younger children and not significantly different between procedures (intravesical (34.6%), liver cyst (25.0%), and choledochal cyst (13.3%)). The authors generally observed low conversion rates, even during the learning phase, and considered RAS to be safe and effective.

A publication from 2013 reported on first experiences with RAS in France [51]. De Lambert and her team evaluated 96 procedures in 94 pediatric patients with an average age of 7.6 years. The study included 57 urologic, 36 general, and 1 thoracic surgery for 12 different procedures. The study demonstrated an average operating time of 189 min and an average hospital stay of 6.4 days (including the day before surgery). Sixteen children showed complications, and three procedures had to be converted (one to open and two to LA). Nonetheless, a success rate of 97% was observed. Moreover, this study provided a listing of the associated costs. The authors calculated the additional cost for RAS compared to LA in terms of fixed costs per hour (of EUR 516) and variable costs per procedure (of EUR 1418). It was concluded that RAS demonstrated good quality results and that it was suitable for implementation in reconstructive surgery.

In 2008, the authors of another study reviewed their first 100 consecutive cases [52]. A total of 89 abdominal (e.g., fundoplication, cholecystectomy) and 11 thoracic procedures (e.g., solid chest tumor surgery) were performed in pediatric patients. The children had a mean age of 8.4 years. The recorded operating time showed great differences and ranged from 53 min (bronchogenic cyst) to over 12 h (total proctocolectomy with ileoanal J-pouch pull-through). The open conversion rates were 12%, with 91.6% of these conversions occurring in abdominal procedures. Although this study underlined the loss of haptic feedback during RAS, the authors stated that RAS was safe and effective across the range of procedures.

## 4. Discussion

A consensus can be discerned across the publications reviewed: the authors of the 36 articles suggest several advantages of RAS over LA and open approaches. Overall, RAS outcomes in children are stated to be feasible, safe, and effective across the range of included procedures (Figure 1). Success rates were high (over 90%), while complication and conversion rates (to LA or open approaches) were acceptably low, with many of the analyzed studies additionally describing RAS as an efficient procedure. As an essential factor for pediatric patients and their parents, the time spent in hospital was assessed in several studies to be shorter compared to other approaches. However, the operating time was significantly longer across diverse publications. This could be explained by the additional docking time required by the robotic system and might be minimized by additional practice and the establishment of appropriate routines. In terms of precision, the minimal invasiveness of RAS makes it a good alternative to LA and open approaches, and RAS was able to outperform both approaches for selected cases and indications. Especially for complex and challenging conditions, the benefits associated with improved visualization, handling, and accuracy further support RAS as a promising approach.

However, a more general patient benefit has not yet been confirmed. This is reflected by major limitations in the current research focusing on RAS in children including the lack of data (beyond the field of urology), as well as the lack of adequate, comparable study designs and application of appropriate statistical methods. Therefore, the present review is limited in its ability to present differentiated outcomes and clear scientific conclusions since it is not possible to do this based on the evidence obtained to date. As highlighted by several authors, support for the future implementation and validation of RAS as an alternative can only be achieved through additional randomized controlled trials.

Furthermore, it should be noted that all the publications investigated in this review reported average ages or bodyweights of patients that were considerably higher than those listed in the inclusion criteria, suggesting that RAS tends to be applied in children who have a higher body weight.

Robotically assisted surgery appears to be applicable to further applications in other surgical fields, but it has several limitations (Figure 2). As noted by multiple authors, there are significant concerns associated with the high cost of RAS compared to LA and open approaches. The purchasing of a robotic system, operating room infrastructure, and payment of maintenance fees represent hurdles. Reusable robotic equipment would be of great benefit with respect to costs and sustainability. It is worth noting that the Senhance^®^ surgical system offers instruments that can be reused, in contrast to the single-use instruments of the da Vinci^®^ surgery system.

With respect to instrument size, 5 mm and 8 mm instruments are often not suitable for small pediatric patients and restrict the application of RAS. In LA, 3 mm instruments represent a safer size with ideal precision and cause minimal tissue damage in pediatric surgeries. Instruments of 3 mm caliber can be placed more closely together while not requiring a long insertional depth. The Senhance^®^ surgical system has recently addressed the crucial role of minimizing the size of the robotic instrumentation for pediatric use. To date, it is the only robotic system providing an adequate portfolio of 3 mm instruments. The smaller size was found to be technically feasible and safe in a piglet study [53]. In addition, some experience with 3 mm instruments in adult gynecological and general surgery has confirmed their feasibility and safety [54]. Nevertheless, the 3 mm instruments that are currently available cannot be angulated. It is necessary to decide individually which specification is more critical for planned procedures—smaller instruments are crucial for surgery in small pediatric patients, while angulation is beneficial for complex reconstructive surgery or extended dissection.

Even though the literature includes exploration of possible limitations and advantages, some practical aspects are hardly addressed. One limitation of RAS might be that the console is usually disconnected from the OR. In emergency or unclear situations, an open and connected console system would seem to have advantages. Regarding the console setup, as addressed by Mattioli and colleagues [41], RAS has a significant ergonomic benefit in terms of a comfortable seating position compared to LA or open approaches, which benefits the surgeon, and, ultimately, the patient. A good ergonomic working situation can contribute to the surgeon being more relaxed and have greater control while operating, resulting in an improved outcome.

As discussed and increasingly evaluated in the literature, the learning curve of RAS and RAS experience play a significant role in its success and observed complication rates, as well as in relation to the operating time. Interestingly, a study of training time involving 42 participants found that less time was taken learning with RAS than with LA [55]. One robotic system, the Senhance^®^ surgical system, implements a new technology, referred to as haptic feedback [4]. This should be beneficial during the surgery itself and should also improve patient safety. Haptic feedback could also help to speed up the learning of surgeons who are new to the robotic system. In addition, compared to LA and open approaches, surgical training on how to use RAS can be provided by video training, which is, arguably, a significant advantage for surgery in children [56].

Considering innovation and technology development, augmented reality (AR) and augmented intelligence might represent the next steps in RAS [1]. AR refers to a mixture of virtual reality and the physical world, where computer-generated input is interposed into the real world [57]. A review by Qian and colleagues inspected AR applications in RAS [58]. AR was used for surgical guidance, interactive surgery planning, port placement, sensory substitution, supervision of robot motion, advanced visualization, bedside assistance, and surgery skill training in an almost exclusively adult population. Surgical guidance, for example, aims to use AR to provide information that is difficult to access intraoperatively or that may be inaccessible to the surgery team. Taken together, AR could add value and safety to RAS in pediatrics in various ways.

## 5. Conclusions

RAS is safe and effective in children. Accordingly, its use has steadily increased across pediatric surgical specialties. With further evaluation of surgical experience and robotic systems, it can be anticipated that its existing limitations will be overcome in the future. For example, some publications focusing on the adult population, and recent case reports relating to the pediatric population, suggest that 3 mm instruments represent a promising technical development in the field of minimally invasive pediatric surgery [45].

Although the authors of this review generally support the more widespread application of RAS, its usage should be considered for selected patient groups in specialized institutions while respecting a precision medicine model.

Finally, the authors strongly believe that the currently available systems and their further development will provide improved and efficient medical care for pediatric patients in diverse subspecialties.

## Figures and Tables

**Figure 1 children-09-00839-f001:**
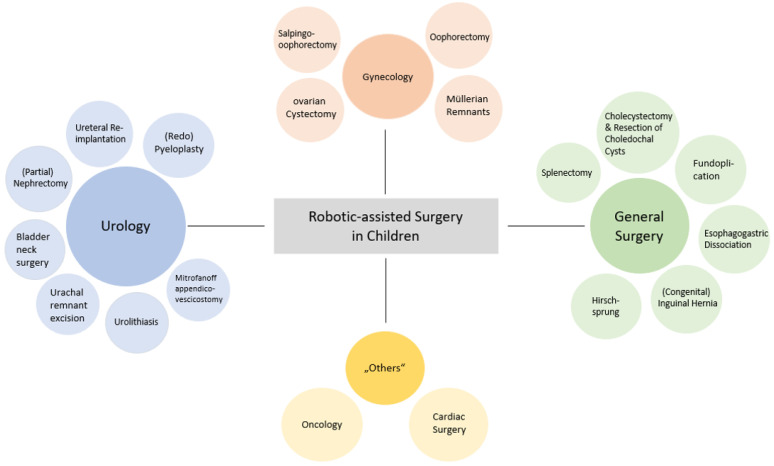
Summarized advantages and limitations of robotically assisted surgery in children.

**Figure 2 children-09-00839-f002:**
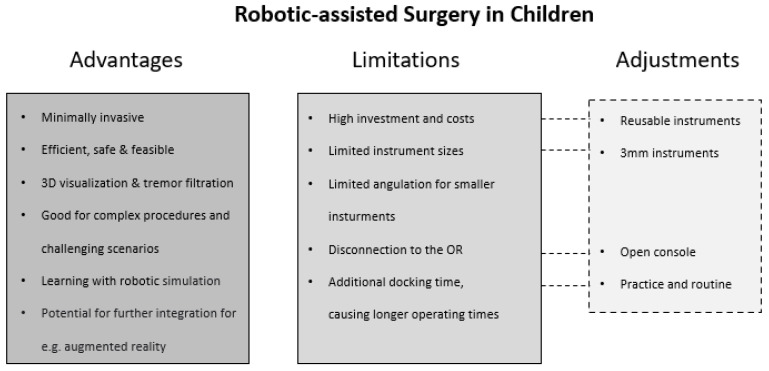
Summarized advantages, limitations and adjustments of robotically assisted surgery in children.

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
