# Peer review of "Robotically Assisted Surgery in Children—A Perspective"

_children, 2022, doi:10.3390/children9060839_

Round 1
Reviewer 1 Report
Very interesting review work, described modern surgical techniques, more and more often used robotic surgery in children. There are no objections to the work. It fully deserves publication.
Author Response
The manuscript was revised on English spelling, grammar, wording, and expression.
Reviewer 2 Report
The author performed a narrative review regarding robotic-assisted surgery in the pediatric population. I read the study with interest. Unfortunately this study lacks in fundamental issues as follow:
- It is unclear what the primary and secondary outcomes of this study were. What exactly did the authors exactly want to investigate? There is just a general statement ‘‘to put current evidence of robotic-assisted surgery in context and highlight further directions’’… but there is not much of that in this text. Text is slightly boring, repeating well-known facts from literature. Also I do not see much regarding further directions in robotic-surgery.
- What parameters / variables were included in analysis, it is too general to search only general terms. There are no key words presented. This should be clearly stated in methodology. Each variable should be analyzed through extracted studies followed by appropriate statistical analysis.
- Why other databases (except PubMed and Google Scolar) were not used for search? At least four databases need to be explored for an efficient search in reviews. (REFERENCE: Bramer WM, Rethlefsen ML, Kleijnen J, Franco OH. Optimal database combinations for literature searches in systematic reviews: a prospective exploratory study Syst Rev. 2017, 6, 245.) Even though this was a narrative review the authors should search more databases.
- The main weakness of this study is that there are no results. None variable has been studied and there is no presentation of the results. Under the chapter ‘results’ there are no results, it looks more like a discussion. This narrative review looks like a chapter from the book. All of this is well-known and reported many times in the literature.
- Also, the report is messy, dealing with various diseases, at the expense of quality. The authors should focus on one disease or group of diseases.
- There is a significant amount of typographic and syntactic errors that need to be revised. The text should be edited by a professional agency or native speaker.
Finally, this report is not structured well, messy, and hard to follow. Unfortunately I do not see any benefits for the readers from this review.
Author Response
- The manuscript was revised on English spelling, grammar, wording, and expression.
- The manuscript structure was further revised and checked on coherence and correctness in results.
1. Aim of the investigation: evaluate studies comparing Robotic-assisted surgery to laparoscopic and open approaches
-
- explore the application of Robotic assisted in other surgery fields besides Urology. What are the outcomes here?
- explore if RAS is a valid alternative to LA and open approaches
2. This work represents a general review, with no aim to be a systematic review. search terms were added.
3. Fair point. two databases were thought two be enough and already provided a vast amount of publications to scan for inclusion.
4. Discussion has been updated on better result reporting. Hopefully, the two figures add some "result" value as well.
5. Quality as been added to each procedure/ indication
6. The manuscript was revised on English spelling, grammar, wording, and expression.
thank you.
Reviewer 3 Report
This short review brings into light the use of RAS in the paediatric population.
Before being considered for publication, I believe that a few aspects need to be adressed.
- The introduction part I believe is too long; you should move several points in the discussion part for more clarity.
- Do not state what was not the aim of the paper, on the contrary state clearly what were the objectives of this review.
- The Methods section should be rephrased using the PICO criteria and the number of papers selected for this review should be given; I would suggest maybe a figure/ flowchart. Give also the number of papers not included in the review, and the reasons why.
- In the discussion part, I believe that a point should be raised regarding the costs of RAS, as compared with laparoscopic or open approach.
I wish the authors the best of luck in their efforts of publishing this review.
Author Response
- Content of introduction was tried to be compromised
- Aim and object are clarified
- We were discussing implementing a PRISMA flowchart. However, this is a general review and no systematic review so we did not see the further need for it.
- The cost discussion part was updated.
Thank you!!!!
Round 2
Reviewer 2 Report
Unfortunately, I do not see any significant improvement in this so-called 'review'. This review has very poor scientific value. The editorial board of Journal Children should decide whether they want state of art publication or average papers of no significant value and very likely no citations.
As I pointed previously there is no novelty from this report. Also the authors mostly ignored my comments.
Methodology is poor, below the standard. Minimum of four databases is mandatory for any serious review article (the authors performed analysis in only two databases although they were advised to perform search of two more databases). Also, there is no date when search was performed. How many investigators performed review of literature, how many articles were identified... Even if this was not a systematic review, the above mentioned data are mandatory.
Also in my previous report I had serious objections regarding chapter 'Results'. I do not see any improvement. The main weakness of this study is that there are no results. None variable has been studied and there is no presentation of the results. Under the chapter ‘results’ there are no results, it looks more like a discussion. This narrative review looks like a chapter from the book. All of this is well-known and reported many times in the literature.
Because of the above mentioned major limitations I can not recommend publication of this article.
Author Response
- There is hardly any recent review that investigated broadly the surgical application of RAS, beyond urology, with an aim to compare RAS to LA and an open approach (Denning et al. 2020, Cave, Simon Clarke 2018). For this work, the additional aim was set to investigate whether RAS is a good alternative approach.
- We searched World of Knowledge and Corchaine for additional reportings. We found 16 additional paper which we screened. Unfuretently they did not meet inclusion cirteria. We added when, how and who the search was performed, including a PRSIMA flow chart.
- We reflected on your comments and decided to rename the manuscript with “perspective”. We agree that it would have been a more meaningful outcome if “variables” (e.g. operating time, conversion rates, etc) were studied and presented individually. But overall, here comes the issue: the majority of studies (especially those that are not urologic procedures) present various study designs with diverse statistical methods. Therefore the current data situation presents a lack of comparable data. This lack complicates the conclusions and thus “scientific value” in this perspective. Taken together, robotic surgery in children is still in its infancy. For this reason, the presentation of our results is only possible in the given way. This work provides readers (surgeons) an overview that RAS is a good alternative to various procedures beyond the field of urology, regarding various variables. It also gives the urgent appeal that future studies with good study design and outcome conclusions are needed.
Thank you very much.
Reviewer 3 Report
No further comments. In my opinion, the introduction part is too long and can be improved, as in my first comment.
Thank you!
Author Response
The manuscript’s introduction was shortened and revised via MPLD English editing check.
Thank you!